# Photoprocesses in Derivatives of 1,4- and 1,3-Diazadistyryldibenzenes

**DOI:** 10.3390/ijms232315346

**Published:** 2022-12-05

**Authors:** Levon S. Atabekyan, Alexandra Y. Freidzon, Dmitry S. Ionov, Vyacheslav N. Nuriev, Alexey V. Medved’ko, Sergey Z. Vatsadze, Sergey P. Gromov, Alexander K. Chibisov

**Affiliations:** 1Photochemistry Center of RAS, FSRC “Crystallography and Photonics”, Russian Academy of Sciences, Novatorov Str. 7A-1, 119421 Moscow, Russia; 2Institute of Nanoengineering in Electronics, Spintronics and Photonics, National Research Nuclear University MEPhI, Kashirskoye Shosse, 31, 115409 Moscow, Russia; 3Department of Chemistry, M. V. Lomonosov Moscow State University, Leninskie Gory 1-3, 119991 Moscow, Russia; 4N.D. Zelinsky Institute of Organic Chemistry RAS, Leninsky Prosp. 47, 119991 Moscow, Russia

**Keywords:** diazadistyrylbenzene, electrocyclization, fluorescence, laser photolysis, triplet state, *trans-cis*-isomerization, density functional theory

## Abstract

Photoprocesses in 1,4-diazadistyrylbenzene (**1**) and 1,3-diazadistyrylbenzene derivative (**2**) diperchlorates in MeCN were studied by absorption, luminescence, and kinetic laser spectroscopies. For compound **1**, *trans-cis*-photoisomerization and intersystem crossing to a triplet state are observed. For compound **2**, photoelectrocyclization is suggested. Quantum chemical calculations of diazadistyrylbenzene structures in the ground and excited states were carried out. The schemes for photoreactions were proposed.

## 1. Introduction

Distyrylbenzenes are promising photoactive compounds for the design of organic light-emitting diodes, materials for solar cells, nonlinear optical materials, and chemical sensors [1,2,3,4,5]. The presence of two C=C double bonds in a molecule of distyrylbenzene offers additional opportunities for their phototransformations. Distyrylbenzenes have been the object of a sufficient number of studies [6,7,8,9,10,11,12,13,14], but there were no data on photoprocesses of diazadistyrylbenzenes and their derivatives. Earlier, we studied the photonics of tetramethoxy-1,4-distyrylbenzenes [15], and also bis-15-crown-5- and 18-crown-6-containing 1,3- and 1,4-distyrylbenzenes [16,17,18,19,20,21] and their ability to form supramolecular complexes with metal cations.

The present study is focused on the derivatives of 1,4- and 1,3-diazadistyrylbenzenes (**1** and **2**, respectively, Figure 1). The aim of this work was a comparative study of the photonics of these compounds in MeCN by absorption, fluorescence, and kinetic laser spectroscopies, detecting the triplet state and intermediate products of photoreactions. In parallel with experimental studies, quantum chemical calculations were carried out to elucidate the structure of the chromophore in the course of photoexcitation, as well as to interpret the experimental absorption and fluorescence spectra.

The established ability of isomeric 1,4- and 1,3-derivatives of distyrylbenzenes to undergo various types of photochemical transformations, including those involving triplet states, has not been studied previously and is of considerable interest. This ability makes them promising for use in photoactive materials. The photoprocesses of diazadistyrylbenzenes under study can also be implemented in the construction of photoactive supramolecular systems on their basis with the use of macrocyclic compounds.

## 2. Results and Discussion

### 2.1. Absorption and Fluorescence Spectra

Figure 2 shows the absorption spectra, and Table 1 represents the corresponding absorption band maxima and extinction coefficients for compounds **1** and **2**. The absorption spectra of **1** and **2** differ both in shape and in the position of the absorption band maxima. A significant blue (40 nm) shift is observed for compound **2** as compared with **1**. In this case, the extinction coefficients of both compounds are virtually identical.

The calculation shows that the absorption of **1** originates from a single delocalized electronic transition with high oscillator strength, while the absorption of **2** originates from four localized electronic transitions with lower oscillator strength, which sum up to almost the same intensity as the absorption of **1**.

Figure 3 shows the fluorescence spectra of compounds **1** and **2**, and Table 2 represents their absorption band maxima, fluorescence quantum yields, and fluorescence lifetimes.

A weak fluorescence of compounds **1** and **2** was detected. Fluorescence excitation spectra coincide with the corresponding absorption spectra. At the same time, compound **2** demonstrates a large Stokes shift of the fluorescence band (109 nm) and a significant (by 2.5 times) decrease in the fluorescence quantum yield compared with compound **1**, which can be explained by a lower degree of conjugation and a significant deformation of the structure upon photoexcitation of **2**. The measured fluorescence lifetime is 120 ps for **1** and 160 ps for **2**. The fluorescence kinetics for compound **2** was found to have a biexponential character, which can confirm the presence of another form of **2** in the excited state with a structure significantly different from the initial one.

The calculated radiative lifetimes are 20–50 times longer than the measured ones, which is consistent with the low fluorescence quantum yield (φ = τ/τ_r_). In the absence of a quencher and the occurrence of the intersystem crossing to the triplet state, the fluorescence lifetime is determined as τ = 1/(*k*_r_ + *k*_nr_ + *k*_ic_). The nonradiative lifetime τ_nr_ for *trans* isomer **1** is 126 ps, for **2,** it is 163 ps, whereas the radiative lifetime is 1420 and 8950 ps, respectively, thus being in agreement with the experimental values.

### 2.2. Kinetic Spectroscopy

Upon laser flash excitation at 355 nm, a change in the absorption spectrum of compound **1** is observed, as illustrated in Figure 4. The corresponding kinetic curves measured in the maxima of the photoinduced absorption bands are shown in Figure 5.

The photoinduced absorbance change at 350–450 nm can be attributed to *trans*-*cis-*isomerization of **1** to give a relatively long-lived *cis-*isomer (Figure 5a). In this case, the rate of decrease in absorption at 400 nm does not depend on the ambient oxygen content in the solution and characterizes the process of *cis-trans* isomerization. The absorption band in the region of 500–650 nm is observed only for the deoxygenated solution and can be associated with the T-T absorption. The triplet state lifetime of **1** is 30 μs (Figure 5b).

The absence of bleaching in the principal absorption band of compound **1** can be attributed to the similarity of the absorption spectra of the *trans* and *cis* isomers of the compound.

For compound **2**, the difference in the photoinduced absorption in the visible spectral range is negligible and more than an order of magnitude smaller than for compound **1**. Figure 6 illustrates the different spectrums of photoinduced absorption of compound **2** under the accumulation of absorption curves.

The observed photoinduced changes in the absorbance of compound **2** can be explained by assuming that these changes occur mainly in the short-wavelength region of the spectrum, which made it impossible to detect them by kinetic laser spectroscopy. In addition to spectral measurements performed upon laser excitation, the absorption spectra of compounds **1** and **2** were measured upon stationary UV irradiation. Figure 7 shows the obtained data.

Upon UV irradiation, an increase in optical density of the solution of compound **1** was observed, which corresponds to a photoinduced change in absorption measured upon laser excitation (Figure 4) and attributed to the formation of a long-lived *cis* isomer of **1**, which has an absorption spectrum similar to that of the *trans* isomer and a significant extinction coefficient.

On the contrary, UV irradiation of compound **2** results in a decrease in absorption of the principal absorption band and an increase in absorption in the short-wavelength region of the spectrum, which is consistent with the experimental data obtained upon laser excitation. It can be assumed that for compound **2**, photoelectrocyclization takes place to give the final product, which absorbs in the short-wavelength region of the spectrum. This conclusion is supported by the literature data confirming the participation of 1,3-distyrylbenzenes in the electrocyclization reaction [1,2,3,8]. Photocyclization of 1,3-diazadistyrylbenzene seems to occur in the singlet excited *cis* isomer, which has a short lifetime. The possible scheme of electrocyclization of compound **2** is depicted in Figure 8.

### 2.3. Quantum Chemical Calculations

#### 2.3.1. Model

In general, the processes of laser flash photolysis experiments can be viewed as follows. 

The reactions of *trans-cis* and *cis-trans* isomerization proceed via a conical intersection (CI), and the minimum energy paths from the local *trans* or *cis* minima on the S_1_ potential energy surface to the CI include saddle points, which limit the photoisomerization rate. The energy of the transition state relative to the *trans* or *cis* local minimum can be considered as the activation energy of *trans-cis* photoisomerization. 

A laser pulse (355 nm) gives the molecule (*trans-***1** or *trans-***2**) excessive energy, which may be sufficient to overcome the barriers of *trans-cis* photoisomerization. After that, the molecules can easily reach the region of the conical intersection and relax nonradiatively either to the *trans* or to the *cis* form in the ground state. This explains why the *trans-cis* isomerization takes place in the nanosecond timescale under the laser flash photolysis conditions.

When photoinduced absorption spectra are measured (after at least 5 μs after the laser pulse), the excess energy is already dissipated either radiatively or nonradiatively, and the mixture consists of the initial molecule in the ground state and its primary photoproducts in their lowest states (S_0_ for singlet products and T_1_ for triplet products). In order to measure photoinduced absorption, the molecules in the mixture are excited directly to their absorption bands without pumping excessive energy. In this case, the barriers for the excited-state reactions can be overcome only through thermal activation, and the reaction rates can be estimated using the Arrhenius formula, given that the corresponding excited state is sufficiently long-lived.

Therefore, after the laser pulse, the mixture consists of both *trans* and *cis* of **1** or **2** in their ground state and, where possible, their triplet forms. Upon measuring the photoinduced absorption spectra, the triplet forms absorb at ~500–600 nm, and the singlet forms absorb at ~350–400 nm, undergo thermally activated or barrierless phototransformations, or emit light (fluoresce). 

#### 2.3.2. Results

The structures of 1,4-diazadistyrylbenzene (**1**) and its electrocyclization product calculated by PBE0/6-31+G(d,p)/PCM(MeCN) are shown in Figure 9. The *trans* isomer is planar both in the ground and in the excited states. The *cis* isomer is nonplanar; the C-C=C-C torsion angle is ~9° in the ground state and ~23° in the excited state. These torsions are close to the same angles in the DHP (~8 and 18° in the ground and excited states, respectively). The ground-state DHP is a local minimum, while the excited-state DHP is a conical intersection (CI). *Trans-cis* photoisomerization proceeds via another conical intersection at ~90° C-C=C-C torsion angle. The minimum energy path from both *trans* and *cis* minima to the CI includes two saddle points at ~121 and 66° C-C=C-C torsion angle.

Figure 10 shows the energy level diagram of diazadistyrylbenzene **1** (a), its possible phototransformations (b), and the absorption maxima of the phototransformation products (c). On the basis of the calculated data, we may infer the following picture of the photoprocesses in **1**.

As mentioned above, after the laser pulse, the mixture consists of *trans-***1**, *cis-***1**, and, possibly, ^3^*trans-***1**. When photoinduced absorption is measured, *trans-***1** absorbs at 420 nm (390 nm in the experiment) with further relaxation to the minimum of the S_1_ state ((*trans*-**1**)*). From the minimum, the molecule can emit light at 455 nm (470 nm in the experiment), convert to the triplet state (^3^*trans-***1**) through S_1_-T_2_ intersystem crossing with further phosphorescence and nonradiative relaxation, or undergo a *trans-cis* isomerization. Without excess excitation energy provided by the laser pulse, this process proceeds in the microsecond timescale and, therefore, is sufficiently slow. The reverse *cis-trans* isomerization in the excited state proceeds much faster, namely, in the nanosecond timescale and can take place even under the conditions of steady-state photoexcitation. Transformation to DHP-**1** from any of the S_1_ minima is unfavorable (denoted by red cross in Figure 10b) because the excited (DHP-**1**)* is ~5.5 kcal/mol higher than the excited (*cis-***1**)* minimum, and the excess energy received by the molecule upon primary flash excitation is dissipated. 

The absorption bands of *cis-***1** and *trans-***1** are very close and overlap when measured (calculated 420 and 412 for *trans-***1** and *cis-***1**, respectively). The emission proceeds in the nanosecond timescale, and the fluorescence spectra of both *cis* and *trans* isomers also overlap (calculated at 455 and 472 nm for *trans-***1** and *cis-***1**, respectively). 

The rate of phototransformation to the triplet state was not evaluated. However, it is worth noticing that T_2_ lies only ~6 kcal/mol lower than S_1_ both in the *trans-* and *cis-* S_1_ minima, and the nature of the T_2_ state is different from the nature of the S_1_ state (see Appendix A). According to the El-Sayed rule, ISC is efficient in this case, leading to further evolution of the triplet molecule. 

Since DHP*-***1** does not accumulate, its oxidation product, phenanthrene-**1** (Phen-**1**), which could be detected by absorption at ~380 nm, does not form.

For the 1,3-diazadistyrylbenzene (**2**) and its electrocyclization product, the corresponding structures calculated by the same method are shown in Figure 11. The *trans* isomer is planar both in the ground and in the excited states. The *cis* isomer is nonplanar; the C-C=C-C torsion angle is ~8° in the ground state and ~14° in the excited state. These torsions are even closer to the same angles in the DHP (~8 and 16° in the ground and excited states, respectively) than in **1**. The ground-state DHP is a local minimum, while the excited-state DHP is a conical intersection. *Trans-cis* photoisomerization proceeds via another conical intersection at ~90° C-C=C-C torsion angle. The minimum energy path from both *trans* and *cis* minima to the CI includes two saddle points at ~128 and 57° C-C=C-C torsion angle.

Figure 12 shows the energy level diagram of diazadistyrylbenzene **2** (a), its possible phototransformations (b), and the absorption maxima of the phototransformation products (c). On the basis of the calculated data, we may infer the following picture of the photoprocesses in **2**.

After the laser pulse, the mixture consists of *trans-***2***, cis-***2**, and, possibly, *^3^trans-***2**. When photoinduced absorption is measured, *trans-***2** absorbs at 314–372 nm (350 nm in the experiment) with further relaxation to the minimum of the S_1_ state ((*trans-***2**)*). From this minimum, the molecule can either emit light at 409 nm (455 nm in the experiment), transform to the triplet state through the S_1_-T_3_ intersystem crossing with further phosphorescence or nonradiative relaxation, or undergo a *trans-cis* isomerization. Without excess excitation energy provided by the laser pulse, this process proceeds in the microsecond timescale and, therefore, is sufficiently slow. The reverse *cis-trans* isomerization proceeds slower by order of magnitude, which may result in the accumulation of the *cis-***2** isomer.

From the (*cis-***2**)* minimum, the molecule can transform to the excited (DHP-**2**)*, which has almost the same energy as the excited (*cis-***2**)* minimum. The (DHP-**2**)* is a conical intersection and serves as a funnel for the nonradiative relaxation of (*cis-***2**)*. The ground-state DHP-**2**, which absorbs in the visible region, is not accumulated due to its fast oxidation to the corresponding phenanthrene-**2** (Phen-**2**), whose absorption band (calculated 358–361 nm) overlaps with those of the *trans-***2** and *cis-***2**.

ISC in the (*trans-***2**)* can proceed from the S_1_ to the closest triplet level, which is T_3_. However, both S_1_ and T_3_ states are formed by the same orbitals (see Appendix A), and this makes ISC inefficient according to the El-Sayed rule (denoted by red cross in Figure 12b). This explains the absence of photoinduced absorption at 500–600 nm in **2**.

The absorption bands of *cis-***2** and *trans-***2** are even closer than in **1** and overlap (calculated 372 and 368 nm for *trans-***2** and *cis-***2**, respectively). The emission proceeds in the nanosecond timescale and the fluorescence spectra of both *cis* and *trans* isomers also overlap (calculated at 409 and 422 nm for *trans-***2** and *cis-***2**, respectively).

## 3. Materials and Methods

The synthesis of perchlorates of 1,4- and 1,3-diazadistyrylbenzenes (**1** and **2**) is beyond the scope of this study and will be published elsewhere. ^1^H-NMR spectra of **1** and **2** can be found in Appendix A.

The absorption spectra were recorded on the Agilent 8453 spectrophotometer. The luminescence spectra were recorded on the Varian Eclipse spectrofluorimeter. Fluorescence lifetimes were measured on the Fluotime 300 spectrofluorimeter. The difference absorption spectra of photoproducts and kinetics decay were measured by means of the nanosecond laser flash photolysis apparatus [22]. Irradiation was performed using the third harmonic of the Nd-YAG laser Solar (*λ* = 355 nm). The dissolved oxygen was removed by the argon bubbling of the solution. Stationary irradiation of diazadistyrylbenzenes solutions was performed using a DKSSh-150 lamp with a UFS-6 glass filter. For the measurement of fluorescence quantum yields, 9,10-diphenylanthracene (quantum yield 0.92 in ethanol [23]) was used as a standard. The accuracy of quantum yield measurement was 10%. All measurements were carried out in MeCN (high purity grade, grade 0, Cryochrome) at room temperature.

Kinetic spectroscopy allows us to draw conclusions about the presence of certain photoreaction products from indirect data (photoinduced absorption spectra and kinetic curves). Since the concentration of the diazadistyrylbenzenes is very low, it is not possible to accumulate and isolate (in general, unstable) photoreaction products in their pure form. 

### Calculation

The structures of the *cis* and *trans* isomers of diazadistyrylbenzenes **1** and **2** and their electrocyclization products (dihydrophenanthrenes DHP*-***1** and DHP*-***2**) in the ground S_0_ and singlet excited S_1_ states, as well as the transition states of the *trans-cis* photoisomerization on the S_1_ potential energy surfaces and DHP oxidation products, phenanthrenes Phen-**1** and Phen-**2**, were optimized by the DFT with PBE0 functional and 6-31+G(d,p) basis set using FireFly program package [24] partially based on GAMESS(US) code [25]. The solvent effect (MeCN) was included through the dielectric polarizable continuum model (D-PCM) [26].

The absorption and emission spectra were calculated by TDDFT with the same functional, basis set, and solvent model. The absorption spectra were calculated after geometry optimization of the ground state, while the emission spectra were calculated after geometry optimization of the π–π* excited state. The vibrational frequencies and normal modes were calculated for all the studied stationary points of the ground and S_1_ excited states. The triplet energies were calculated by TDA [27].

The radiative lifetime *τ_r_* was calculated as follows:(1)$$k_{r}=\frac{2}{3}f_{0i}\nu_{i0}^{2};\;\;\;\;\;\;\;\;\;\;\;\;\;\;\;\;\tau_{r}=\frac{1}{k_{r}}$$
where *k_r_* is the fluorescence rate constant, *f*_0*i*_ is the oscillator strength, and ν*_i_*_0_ is the frequency (cm^−1^) of the transition.

The *trans-cis* and *cis-trans* photoisomerization times *τ_tc_* were calculated using the Arrhenius equation:(2)$$\;k_{tc}=c\omega_{0}\exp\left(\frac{-E_{A}}{kT}\right);\;\;\;\;\;\;\;\;\;\;\;\;\;\;\;\;\;\tau_{tc}=1/k_{tc}$$
where *E_A_* is the activation energy calculated for the corresponding transition state on the S_1_ potential energy surface, *c* is the speed of light, and *ω*_0_ is the frequency of the reorganizational mode [28,29,30].

Since the energy profile of the isomerization and subsequent electrocyclization goes through conical intersection regions, the applicability of TDDFT needs special discussion. It is known that TDDFT is inapplicable to cases with quasidegeneracy. In the quasidegeneracy region, the excited state geometry optimization procedure fails because the gap between the ground and excited state (or the excitation frequency) becomes zero, and Casida equations do not have positive eigenvalues [31]. However, for our qualitative conclusions, we neither need the exact energy of the minimum energy CI nor its exact structure. For our purposes, it is sufficient to know that both (DHP-**1**)* and (DHP-**2**)* lie in the region of the conical intersection. If this intersection is peaked [32], it acts as a reaction funnel. By TDDFT, we may roughly demonstrate that the near-CI region for (DHP-**1**)* lies higher than the local *cis* and *trans* minima of the S_1_ state, while for (DHP-**2**)*, this region has lower energy and, therefore, a barrierless path from the *cis* form to the CI is possible in **2**.

The calculations were also repeated using BHandHLYP functional to demonstrate that the results are qualitatively the same and only slightly depend on the weight of exact exchange in the functional (Appendix A)

This methodology has been benchmarked against multireference (XMCQDPT2/CASSCF) calculation in [33].

## 4. Conclusions

It was found that the photoprocesses of 1,4- and 1,3-isomers of diazadistyrylbenzenes differ substantially. The phototransformation of the 1,4-diazadistyrylbenzene derivative involves *trans-cis* isomerization and intersystem crossing to the triplet state. In this case, the formation of DHP-**1** is energetically unfavorable. For 1,3-diazadistyrylbenzene, *trans*-*cis* isomerization is followed by the electrocyclization from the *cis-*form to generate DHP-**2** and further fast oxidation with atmospheric oxygen to give phenanthrene, while intersystem crossing is not efficient. The considered photoprocesses of diazadistyrylbenzenes can be implemented in the construction of photoactive supramolecular systems on their basis with the use of macrocyclic compounds.

## Figures and Tables

**Figure 1 ijms-23-15346-f001:**
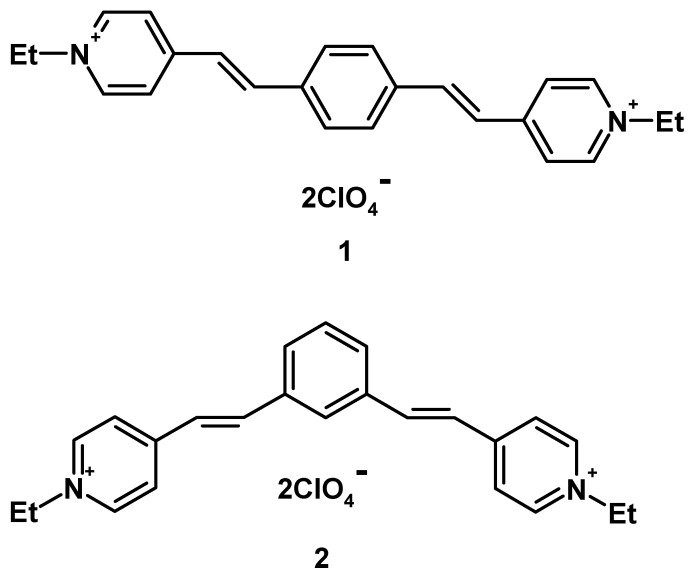
Structures of 4,4’-{1,4-phenylenebis[(E)-ethene-2,1-diyl]}bis(1-ethylpyridinium diperchlorate (**1**) and 4,4’-{1,3-phenylenebis[(E)-ethene-2,1-diyl]}bis(1-ethylpyridinium) diperchlorate (**2**).

**Figure 2 ijms-23-15346-f002:**
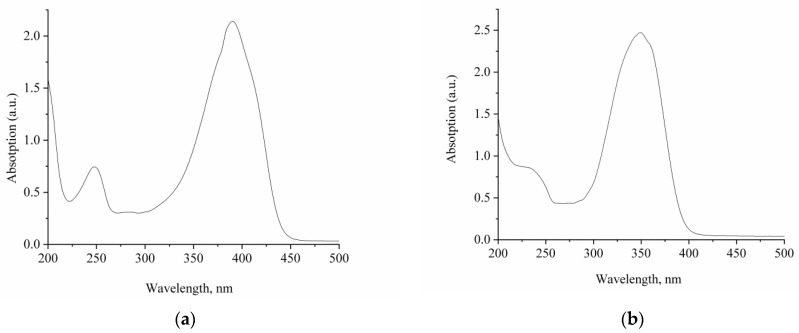
Absorption spectra of compounds (**a**) **1** and (**b**) **2**. The concentration of compounds is 4 × 10^−5^ (**a**) and 5 × 10^−5^ (**b**) mol/L.

**Figure 3 ijms-23-15346-f003:**
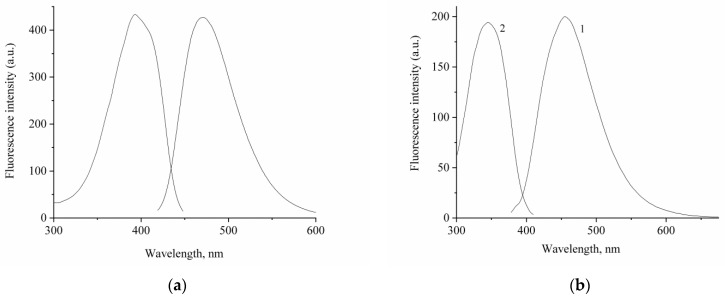
Fluorescence emission (1) and excitation (2) spectra of compounds (**a**) **1** and (**b**) **2**. The concentration of compounds **1** and **2** is 5 × 10^−6^ mol/L.

**Figure 4 ijms-23-15346-f004:**
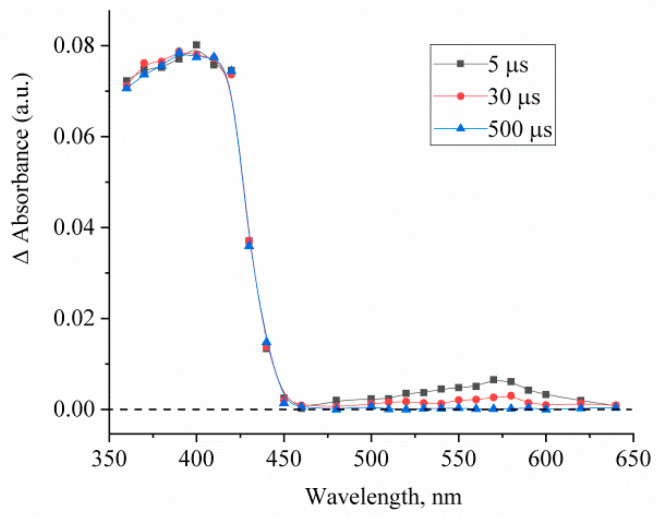
Difference photoinduced absorption spectra of deoxygenated solution of compound **1** measured at 5, 30, and 300 μs after the laser pulse. The concentration of compound **1** is 5 × 10^−5^ mol/L.

**Figure 5 ijms-23-15346-f005:**
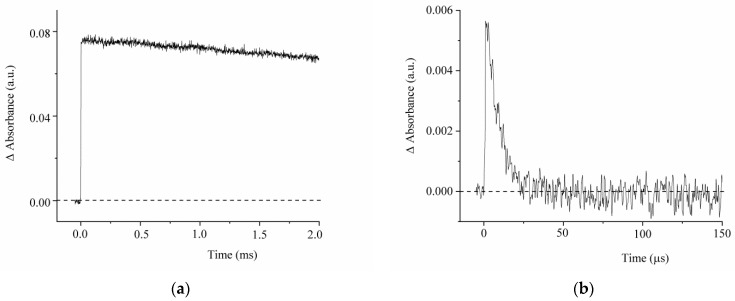
Kinetic curves of the absorbance change in the deoxygenated solution of compound **1** measured at λ = 400 nm (**a**) and 570 nm (**b**). The concentration of compound **1** was 5 × 10^−5^ mol/L.

**Figure 6 ijms-23-15346-f006:**
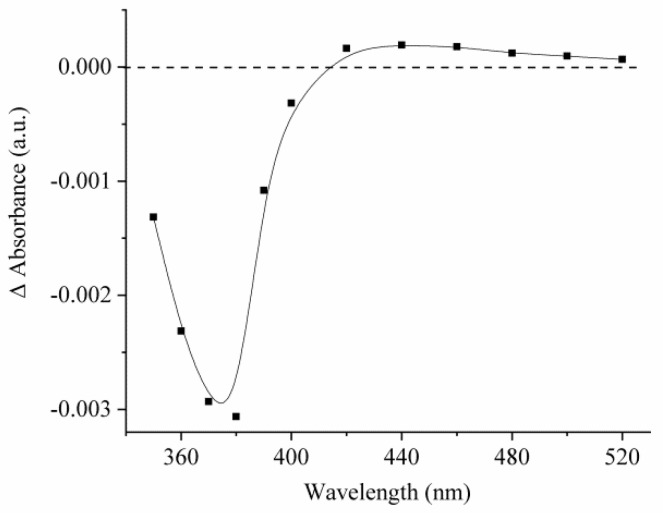
Photoinduced absorption spectrum of compound **2** measured at 5 μs after the laser pulse. The concentration of compound **2** was 5 × 10^−5^ mol/L.

**Figure 7 ijms-23-15346-f007:**
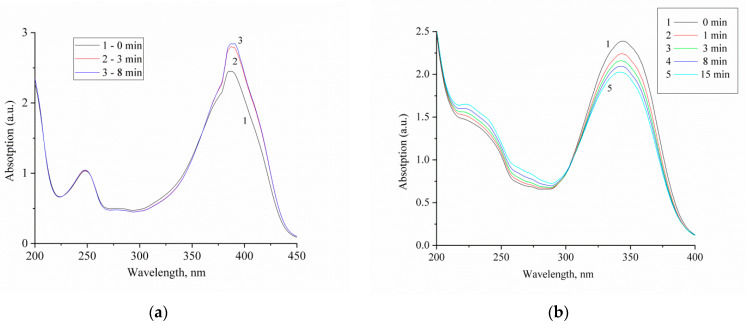
Absorption spectra of compounds **1** (**a**) and **2** (**b**) measured upon UV irradiation at various times as indicated in the insets. The concentration of compounds **1** and **2** was 5 × 10^−5^ mol/L.

**Figure 8 ijms-23-15346-f008:**
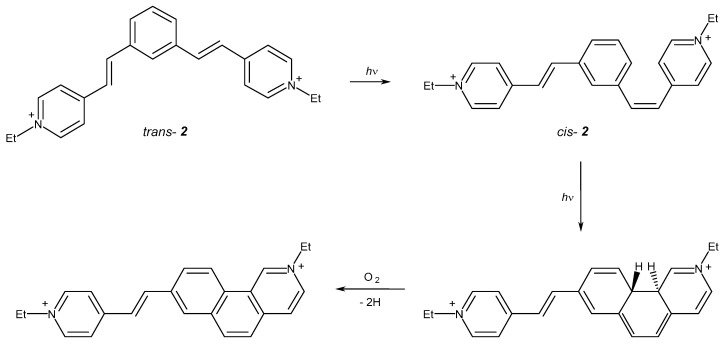
Scheme of electrocyclization of compound **2**.

**Figure 9 ijms-23-15346-f009:**
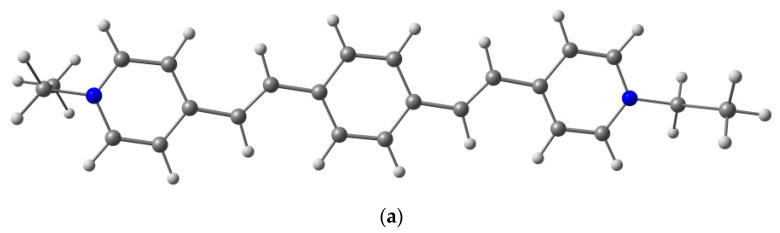
Structures of (**a**) *trans-***1** and (**b**) *cis-***1** and (**c**) its electrocyclization product dihydrophenanthrene DHP*-***1**.

**Figure 10 ijms-23-15346-f010:**
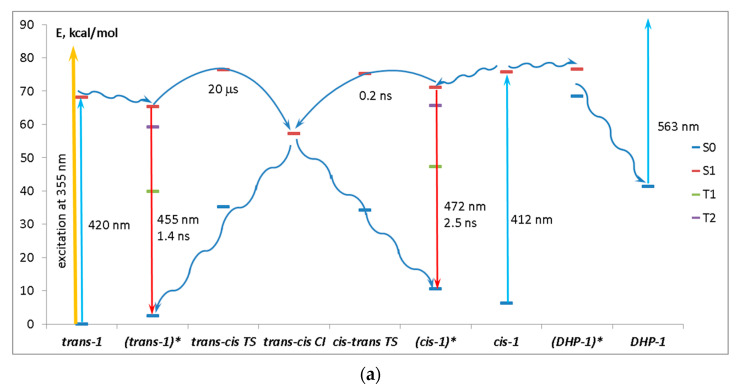
(**a**) Energy level diagram of diazadistyryl benzene **1**, (**b**) scheme of its possible phototransformations, and (**c**) absorption maxima of the phototransformation products. Straight arrows on the energy diagram denote radiative processes, and wavy arrows denote nonradiative processes.

**Figure 11 ijms-23-15346-f011:**
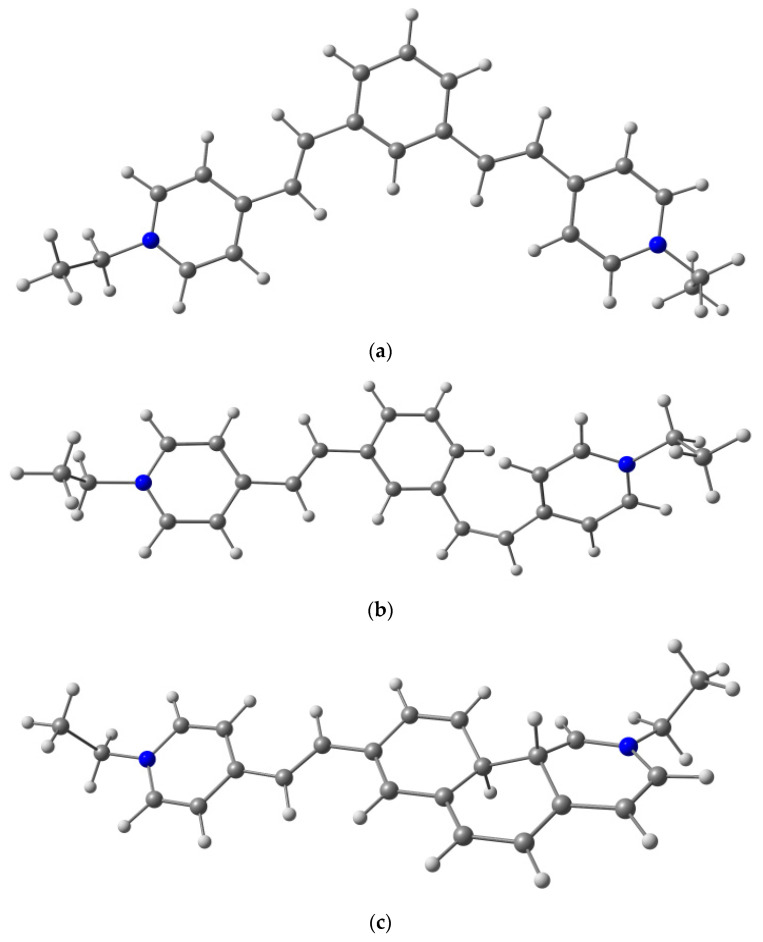
Structures of (**a**) *trans-***2** and (**b**) *cis-***2** and (**c**) its electrocyclization product dihydrophenanthrene DHP*-***2**.

**Figure 12 ijms-23-15346-f012:**
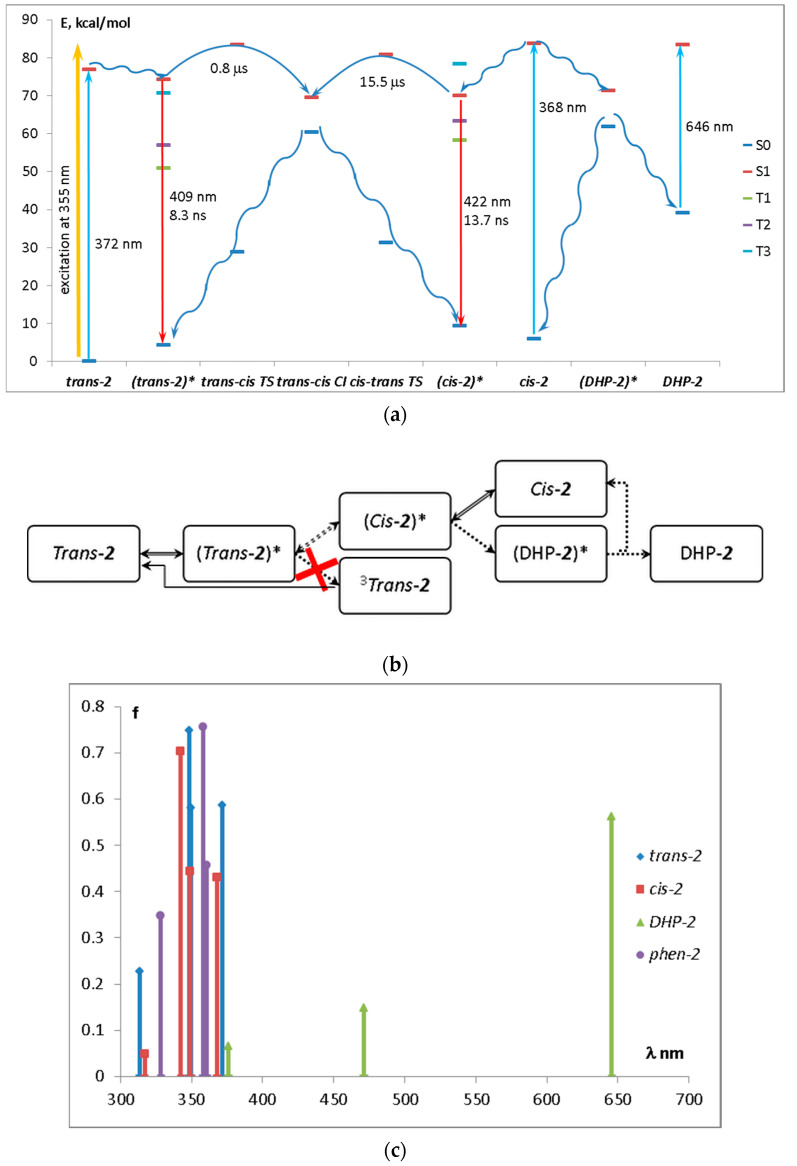
(**a**) Energy level diagram of diazadistyrylbenzene **2**, (**b**) scheme of its possible phototransformations, and (**c**) absorption maxima of the phototransformation products. Straight arrows on the energy diagram denote radiative processes, whereas wavy arrows denote nonradiative processes.

**Table 1 ijms-23-15346-t001:** Absorption band maxima and extinction coefficients.

Compound	Experiment		Calculation	
	λ_max_, nm	ε, L∙mol^−1^∙cm^−1^	λ_max_, nm	*f*
1	390	5.4 × 10^4^	420	2.07
2	350	5.0 × 10^4^	372 ^1^ 350 ^1^ 348 ^1^ 314 ^1^	0.59 ^1^ 0.58 ^1^ 0.75 ^1^ 0.23 ^1^

*f* is the oscillator strength; ^1^ four absorption bands of **2**.

**Table 2 ijms-23-15346-t002:** Emission band maxima, fluorescence quantum yields φ and fluorescence lifetimes.

Compound	Experiment						Calculation		
	φ	λ_max_, nm	λ_0-0_, nm	λ_exc_, nm	Δλ, nm	τ_fl_, ps	λ_max_, nm	*f*	τ_r_, ps
1	0.05	470	434	393	77	120	454 ^1^ 472 ^2^	2.18 1.34	1420 2500
2	0.02	455	395	346	109	160	409 ^1^ 422 ^2^	0.30 0.19	8350 13700

Δλ is a Stokes shift between the fluorescence emission and excitation maxima; ^1^ *trans* isomer; ^2^ *cis* isomer.

## Data Availability

Not applicable.

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
