# Peer review of "Photoprocesses in Derivatives of 1,4- and 1,3-Diazadistyryldibenzenes"

_ijms, 2022, doi:10.3390/ijms232315346_

Round 1

Reviewer 1 Report

The authors report on the concurrent study of the title compounds experimentally and theoretically.  The scope of this study matches the policy of IJMS, and it includes huge amount of data and calculations which are presented in a consistent manner.  The reviewer would like to recommend the manuscript for acceptance with minor revision.  Each comment is listed below.

(1) As written in the author's guide, references should be written in Arabic numerics such as "1, 2", not as "i and ii".  Consecutive references should be written as "[1-5]".

(2) L55: should be written as "compounds 1 and 2" (some character was erroneously inserted).

(3) There are two references with missing last page numbers (ix and x), and they should be corrected.

Author Response

We thank the reviewer for carefully reading the manuscript.

(1) As written in the author's guide, references should be written in Arabic numerics such as "1, 2", not as "i and ii".  Consecutive references should be written as "[1-5]".

Corrected

(2) L55: should be written as "compounds 1 and 2" (some character was erroneously inserted).

Corrected

(3) There are two references with missing last page numbers (ix and x), and they should be corrected.

Corrected

Reviewer 2 Report

The paper by Atabekyan et al reports on the luminescent properties and photochemical transformation of two isomeric forms of distiryl-dibenzenes.

I find the paper technically sound and the results of interest to the readership of IJMS. I recommend its publication with only a minor revision:

The lambdas in Table 1 have Cyrillic characters that are probably typos. The same for line 55.

Author Response

We thank the reviewer for carefully reading the manuscript.

The lambdas in Table 1 have Cyrillic characters that are probably typos. The same for line 55.

Corrected

Reviewer 3 Report

This work presents a study of photo-properties and -behavior of 1,4-diazadistyrylbenzene and 1,3-diazadistyrylbenzene. This manuscript has been competently done and it is technically correct. On the other hand, its novelty and significance for the scientific community is questionable. Therefore, I would recommend rejection and submission to a more specialized journal.

Author Response

This work presents a study of photo-properties and -behavior of 1,4-diazadistyrylbenzene and 1,3-diazadistyrylbenzene. This manuscript has been competently done and it is technically correct. On the other hand, its novelty and significance for the scientific community is questionable. Therefore, I would recommend rejection and submission to a more specialized journal.

The established ability of isomeric 1,4- and 1,3-derivatives of distyrylbenzenes to undergo various types of photochemical transformations, including those involving triplet states, has not been studied previously and is of considerable interest. This ability makes them promising for use in photoactive materials. The considered photoprocesses of diazadistyrylbenzenes can also be implemented in the construction of photoactive supramolecular systems on their basis with the use of macrocyclic compounds.

Reviewer 4 Report

The paper by Atabekyan et al. reports the photophysical and computational study of the behaviour of two compounds. The formal aspects of the manuscript are adequate. The contents are within the scope of the journal.

The paper is divided into two main parts. One is focused on the photophysical properties of the compounds in which absorption and fluorescence are measured and discussed. The second part tries to explain the results using computational methodology. In both sections, a comparison between the two studied compounds is presented. While the first part is incomplete but reasonable, the second one has some limitations.

Regarding the experimental part, some details simply cannot be accepted. No preparation or characterization of the compounds is presented. The sentence "The synthesis of perchlorates of 1,4- and 1,3-diazadistyrylbenzenes (1 and 2) will be published elsewhere." (l198) cannot hide the fact that there is no information on the preparation and purity of the compounds. Beyond obvious fragmentation of the work, this implies that this work cannot be reproduced and this should not be allowed. In a more general way, one can argue why the authors have decided to infer the formation of the photoproducts from indirect methods instead of isolating them. Photolysis of 1 and 2 should yield different results and this could be used as direct proof of the transformation proposed.

The computational part section has also some problems. No Cartesian coordinates are provided, the methodology has not been benchmarked and, thus, results are just approximate. Figures with the computed structures are just poured without a minimal geometry discussion. The sentence (l187) "The (DHP-2)* is a conical intersection and serves as a funnel for the nonradiative" is directly wrong as DFT cannot represent faithfully a conical intersection seam. The ISC points are suggested but not computed. The computed trans-to-cis TS structures are possibly meaningless as the photoreaction typically proceeds through a conical intersection. The sentence (l152) "Transformation to DHP-1 from the S1 minimum is unfavorable, because excited (DHP-1)* is ~6 kcal/mol higher than the excite" is also meaning as this transformation could also proceed through a conical intersection.

In general, the paper has some serious drawbacks that prevent publication in its present form. I would suggest the authors to further elaborate on both sections in the manuscript with additional data and a proper discussion of the results.

Author Response

Regarding the experimental part, some details simply cannot be accepted. No preparation or characterization of the compounds is presented. The sentence "The synthesis of perchlorates of 1,4- and 1,3-diazadistyrylbenzenes (1 and 2) will be published elsewhere." (l198) cannot hide the fact that there is no information on the preparation and purity of the compounds. Beyond obvious fragmentation of the work, this implies that this work cannot be reproduced and this should not be allowed. 

NMR spectra of title compounds 1 and 2 were added to Supporting Information to prove their purity. The synthesis of the diazadistyrylbenzenes is beyond the scope of this paper and is not given here.

In a more general way, one can argue why the authors have decided to infer the formation of the photoproducts from indirect methods instead of isolating them. Photolysis of 1 and 2 should yield different results and this could be used as direct proof of the transformation proposed.

Kinetic spectroscopy allows us to draw conclusions about the presence of certain photoreaction products from indirect data (photoinduced absorption spectra and kinetic curves). Since the concentration of the diazadistyrylbenzenes is very low, it is not possible to accumulate and isolate (in generally, unstable) photoreaction products in their pure form.

The computational part section has also some problems. No Cartesian coordinates are provided, the methodology has not been benchmarked and, thus, results are just approximate.

We thank the reviewer for valuable comments. It helped us improve the computational section of the paper.  We added Cartesian coordinates of all the structures to Supporting Information.

Figures with the computed structures are just poured without a minimal geometry discussion.

We added geometry discussion.

The sentence (l187) "The (DHP-2)* is a conical intersection and serves as a funnel for the nonradiative" is directly wrong as DFT cannot represent faithfully a conical intersection seam. 

Indeed, DFT is not appropriate to locate conical intersections or other regions with quasidegeneracy. 

In the quasidegeneracy region the excited state geometry optimization procedure fails, because the gap between the ground and excited state (or the excitation frequency) becomes zero, and Casida equations do not have positive eigenvalues [[i]]. However, for our qualitative conclusions we neither need the exact energy of the minimum energy CI nor its exact structure. For our purposes it is sufficient to know that both (DHP-1)* and (DHP-2)* lie in the region of the conical intersection. If this intersection is peaked [[ii]], it acts as a reaction funnel. By TDDFT we may roughly demonstrate that the near-CI region for (DHP-1)* lies higher than the local cis and trans minima of the S1 state, while for (DHP-2)* this region has lower energy and, therefore, a barrierless path from the cis form to the CI is possible in 2.

This methodology has been benchmarked against multireference (XMCQDPT2/CASSCF) calculation in [[iii]].

[i] Casida M. Time-Dependent Density Functional Response Theory for Molecules in Recent Advances in Density Functional Methods, Chong, D.P., Ed. World Scientific, Singapore, 1995, p.155-192. https://doi.org/10.1142/9789812830586_0005

[ii] Atchity, G.J.; Xantheas, S.S.; Ruedenberg, K. Potential energy surfaces near intersections, J. Chem. Phys. 1991, 95, 1862–1876. https://doi.org/10.1063/1.461036

[iii] Quentin, C.; Gerasimaitė, R.; Freidzon, A.; Atabekyan, L.S.; Lukinavičius, G.; Belov, V.N.; Mitronova, G.Y. Direct Visualization of Amlodipine Intervention into Living Cells by Means of Fluorescence Microscopy, Molecules 2021, 26(10), 2997; https://doi.org/10.3390/molecules26102997.

The ISC points are suggested but not computed.

ISC points were not calculated because DFT also performs poorly for triplet states, and this issue cannot be circumvented within this study. Nevertheless, we give the triplet energies in the cis and trans minima of the S1 state just to demonstrate the possibility of ISC in these states. However, we do not give any rate constants based on SOC matrix elements, because they are unreliable for DFT-calculated states.

The computed trans-to-cis TS structures are possibly meaningless as the photoreaction typically proceeds through a conical intersection. 

We thank the reviewer for drawing our attention to the trans-cis isomerization part. Indeed, careful investigation allowed us to build a correct energy diagram including both a trans-cis CI (again, approximate, as in the case of DHP) and two transition states from the trans and cis minima to the CI. Since crossing the barrier limits the trans-cis isomerization rate, we estimated the rate using the Arrhenius equation.

The calculated isomerization times lie in the microsecond scale, which conflicts with the fact that primary photoproducts, including cis isomer, are formed during the laser pulse (nanosecond timescale). Laser pulse excitation at 355 nm provides sufficient excess energy to overcome the barrier non-thermally, which makes Arrhenius equation inapplicable. Therefore, the calculated isomerization rates are applicable only to the excitation directly to the absorption band of either trans or cis form, which takes place upon measurement of photoinduced absorption after the laser pulse.

The sentence (l152) "Transformation to DHP-1 from the S1 minimum is unfavorable, because excited (DHP-1)* is ~6 kcal/mol higher than the excite" is also meaning as this transformation could also proceed through a conical intersection.

One of the most important results of our study is to show that the energy minimization of the (DHP-1)* leads to a CI, and this CI lies higher than both trans and cis minima on the S1 potential energy surface. It is not a peaked CI; therefore, getting to this region needs some activation. At the same time, other processes in compound 1 are faster and successfully compete with electrocyclization. In (DHP-2)* energy minimization also leads to a CI, but it lies below the trans and cis minima on the S1 potential energy surface and, therefore, can be a funnel.

Round 2

Reviewer 3 Report

The revised version has been improved, in terms of clarity. 

As I wrote before, the work has been competently done, and my only 

concern was how this topic will be interesting for general chemical audience.

Author Response

As I wrote before, the work has been competently done, and my only concern was how this topic will be interesting for general chemical audience.

We thank the reviewer. We believe that the behavior of dyes upon laser excitation may be of interest for the design of photoactive materials, such as recording media, sensors, etc.

Reviewer 4 Report

The authors have made some efforts to improve the manuscript in this version. However, most of the changes imply minor modifications, and the critical flaws have not been addressed.

- The authors consider that the synthesis is out of the scope of the manuscript. I do not challenge this vision. However, the results shown here cannot be validated without access to the materials. Again, this is an evident work fragmentation. If the authors prefer to do so, I would suggest going with the synthesis first.

- I agree with the authors that some intermediates could be hard to isolate. However, the photoisomer and, especially, the electrocyclized compound, should be easy to isolate. The authors' claim on low concentration suggests difficult access to starting materials. Again, the synthetic route is relevant here.

- The computational methodology benchmark I was suggesting referred to the different functional and /or basis sets. The poor performance of DFT vs. CASSCF in this type of photoreaction is clear.

- I agree with the fact that some qualitative information can be obtained from TD-DFT. However, labelling some points as CI using TD-DFT is wrong.  How are those points computed? Probably through an excited state minimization which is not a CI searching algorithm. In addition, the CI point topology cannot be explored with TD-DFT.

-Some conclusions, even if probable, are supported by very fragile data. If a qualitative theory is used, whatever the reason, conclusions should be, at a maximum, at the same level of confidence. This is the main problem with the paper which has not been corrected in this revision.

Author Response

The authors have made some efforts to improve the manuscript in this version. However, most of the changes imply minor modifications, and the critical flaws have not been addressed.

- The authors consider that the synthesis is out of the scope of the manuscript. I do not challenge this vision. However, the results shown here cannot be validated without access to the materials. Again, this is an evident work fragmentation. If the authors prefer to do so, I would suggest going with the synthesis first.

We cannot agree with the reviewer that the NMR data presented are insufficient to assess the purity and confirm the structure of the compounds in study. The synthesis involves many stages and is comparable in volume to the presented work. It cannot be included in the current paper; instead, it deserves a dedicated paper and will be published later. 

- I agree with the authors that some intermediates could be hard to isolate. However, the photoisomer and, especially, the electrocyclized compound, should be easy to isolate. The authors' claim on low concentration suggests difficult access to starting materials. Again, the synthetic route is relevant here.

Both compounds are available in the amounts sufficient for the study; however, the conditions of kinetic laser spectroscopy experiments do not allow one to accumulate the photoreaction products.

- The computational methodology benchmark I was suggesting referred to the different functional and /or basis sets. The poor performance of DFT vs. CASSCF in this type of photoreaction is clear.

Actually, poor performance of DFT in CI regions is a well-known fact. It does not depend on the functional. We repeated the same calculations by BHHLYP (energy diagrams added to SI), and the picture is qualitatively the same, except for the fact that excitation and emission energies are overestimated. Such large difference between the experimental and calculated values makes interpretation of the experiment with the use of BHHLYP data difficult. This is why we use PBE0 results for interpretation.

- I agree with the fact that some qualitative information can be obtained from TD-DFT. However, labelling some points as CI using TD-DFT is wrong.  How are those points computed? Probably through an excited state minimization which is not a CI searching algorithm. In addition, the CI point topology cannot be explored with TD-DFT.

The near-CI regions (actually, not the MECI points) were determined as the points where excited state optimization stopped because the A-B matrix is not positively definite. In the current revision of the manuscript, we explain this explicitly. We cannot guarantee that the geometry found is a MECI, but we hope it is sufficiently close to it. The S0-S1 gap in this geometry is small enough to state this.

-Some conclusions, even if probable, are supported by very fragile data. If a qualitative theory is used, whatever the reason, conclusions should be, at a maximum, at the same level of confidence. This is the main problem with the paper which has not been corrected in this revision.

Our conclusions are qualitative: we demonstrate that the most probable relaxation channel in compound 1 is ISC with further evolution of the triplet state, while in compound 2 it is isomerization with further electrocyclization. We evaluate the reaction rates only to show the timescale of the processes involved. Our conclusions agree with the observed kinetic laser spectroscopy data.

Round 3

Reviewer 4 Report

The authors have not treated this revision with care, as just a few suggestions have been considered in the last version. My opinion for this manuscript stands as they authors have not modified any of the critical points, not provided additional evidence to induce a change in my views. From the last response:

-"We cannot agree with the reviewer that the NMR data presented are insufficient to assess the purity and confirm the structure of the compounds in study. The synthesis involves many stages and is comparable in volume to the presented work. It cannot be included in the current paper; instead, it deserves a dedicated paper and will be published later. "

My critic was never directed to the poor characterization, but to the fact that not including the synthesis in this paper prevents the replication of the experiments. This could be easily sorted out by publishing the synthesis first.

-"The near-CI regions (actually, not the MECI points) were determined as the points where excited state optimization stopped because the A-B matrix is not positively definite. In the current revision of the manuscript, we explain this explicitly. We cannot guarantee that the geometry found is a MECI, but we hope it is sufficiently close to it. The S0-S1 gap in this geometry is small enough to state this."

Those points are meaningless. And still they labeleled as MECI. We cannot base our conclusions in "hope". In case of a sloped CI, conclusions will not be valid. And the used methodology simply cannot assess that.